# DM2C: Deep Mixed-Modal Clustering

**Yangbangyan Jiang**[1,2]     **Qianqian Xu**[3]     **Zhiyong Yang**[1,2]
**Xiaochun Cao**[1,2,6]     **Qingming Huang**[3,4,5,6*]

[1]State Key Laboratory of Information Security, Institute of Information Engineering, CAS
[2]School of Cyber Security, University of Chinese Academy of Sciences
[3]Key Lab. of Intelligent Information Processing, Institute of Computing Technology, CAS
[4]School of Computer Science and Tech., University of Chinese Academy of Sciences
[5]Key Laboratory of Big Data Mining and Knowledge Management, CAS
[6]Peng Cheng Laboratory
jiangyangbangyan@iie.ac.cn, xuqianqian@ict.ac.cn, yangzhiyong@iie.ac.cn
caoxiaochun@iie.ac.cn, qmhuang@ucas.ac.cn

## Abstract

Data exhibited with multiple modalities are ubiquitous in real-world clustering tasks. Most existing methods, however, pose a strong assumption that the pairing information for modalities is available for all instances. In this paper, we consider a more challenging task where each instance is represented in only one modality, which we call *mixed-modal data*. Without any extra pairing supervision across modalities, it is difficult to find a universal semantic space for all of them. To tackle this problem, we present an adversarial learning framework for clustering with mixed-modal data. Instead of transforming all the samples into a joint modality-independent space, our framework learns the mappings across individual modality spaces by virtue of cycle-consistency. Through these mappings, we could easily unify all the samples into a single modality space and perform the clustering. Evaluations on several real-world mixed-modal datasets could demonstrate the superiority of our proposed framework.

## 1 Introduction

Recently supervised classification tasks have achieved impressive performance with the development of deep learning. However, such improvement often relies on a large number of manual annotations which are very expensive and laborious. On the contrary, unsupervised clustering remains an appealing direction for deep learning since it works in the absence of data labels.

Various efforts have been devoted to addressing the problem of partitioning data in a single modal form [32, 14, 17, 18]. Yet real-world data are often characterized by multiple modalities. For example, a data object (say *a web page* or *a node in the social network*) can be exhibited by both visual images and text tags/captions. Learning with multiple modalities offers us a chance to reach a thorough comprehension on the data by means of integrating modality-specific information coming from each modality. Therefore, clustering multi-modal data has become an active research area in recent years [5, 12, 7]. The key problem of this task is how to learn a joint representation for each sample against the semantic gap across modalities. Most existing work tries to find a solution under an ideal assumption that each modality is available for all the samples. This, however, requires gross human efforts on data collection since real-world data often suffer from some missing information for each modality. In the worst case, when the semantic connection across modalities is completely missing, we come to find our samples represented in one modality, *e.g.*, a twitter post may only include either

---

[*]Corresponding author.

Table 1: Types of learning under multiple modalities.

| Type | Supervision | |
|------|-------------|--|
| | Class Label | Modality Pairing |
| Supervised Multi-modal Learning | ✓ | ✓ |
| Unsupervised Multi-modal Learning | ✗ | ✓ |
| Unsupervised Mixed-modal Learning | ✗ | ✗ |

an image or a text. How to deploy clustering in this case is still a puzzle to our community. In this paper, we focus on the clustering problem with such worse case where each sample only consists of one of several modalities, *i.e.*, **mixed-modal data**. We assume that there exists an underlying relationship among the modalities and then seek an algorithm by exploiting such a relationship. At first glance, one might resort to learning a joint space for the features extracted for each modality. Actually, this is the solution widely adopted in traditional multi-modal clustering methods, when the pairing information across modalities for each sample is available. However, this is no longer suitable for the mixed-modal setting. As summarized in Table 1, in a unsupervised mixed-modal setting, the model is learned without any form of supervision including modality pairing information. Hence it is hard to find the correlation across different modalities, let alone the alignment. In such a case, transforming all the samples into a joint semantic space is almost impossible.

Meanwhile, Generative Adversarial Network (GAN), especially CycleGAN [39], has become an effective means of dealing with unsupervised learning for data across multiple modalities or domains. In CycleGAN, a *cycle-consistency* constraint is proposed to enforce the connection across domains, where translating a sample from domain $A$ to $B$ and then reconstructing it back to $A$ should result in the original sample representation. This framework has shown a great power to build the mappings for unpaired data [16, 23, 33]. Inspired by this, we turn to learn the translation across different modalities, by which we can unify the representations into one modality and perform the clustering.

Specifically, we propose an adversarial learning framework to tackle the unified representation learning for mixed-modal clustering task. The key idea of our framework lies in that cross-modal generators are implemented to learn the bidirectional mappings between modalities via the cycle consistency constraint, while modality-specific discriminators try to distinguish between data in a specific modality and transformed from other modalities. To do this, we first reconstruct data using the corresponding modality-specific auto-encoders to obtain the latent representations. Then a cycle-consistent mini-max game is performed on the discriminators and the mappings between modalities. Equipped with the unified representations, a common clustering algorithm is performed to get the final results. Experimental results on real-world mixed-modal datasets show that the proposed cycle-consistent framework obtains better performance than the competitors.

This paper is organized as follows. First, we briefly review the recent development for the related areas in Section 2. Next, we detail the formulation for the mixed-modal setting and our proposed method in Section 3. Then we evaluate our performance on several real-world mixed-modal datasets in Section 4. Finally, in Section 5, we give the concluding remarks regarding the mixed-modal clustering problem.

## 2  Related work

**Multi-modal/view clustering.**    Traditional multi-modal/view clustering aims at grouping objects which have different representations in different modalities/views. A typical strategy to bridge the disjoint feature spaces is to co-regularize the representations/structures for all the modalities/views. For example, Canonical Correlation Analysis projects all the samples onto a latent shared subspace by maximizing the correlations among instances in different feature spaces [7, 6]; multi-view spectral clustering constructs a common transition probability matrix [38, 5, 40]; multi-view subspace clustering aggregates the subspace structure [12, 21, 35, 34]; multi-view Non-negative Matrix Factorization calculates a consensus coefficient matrix [13, 41]. Although these approaches achieve very promising performance, they require that all the samples are exhibited in all the modalities/views. Considering the lack of pairing information, partial multi-view clustering is proposed for the condition where some views are missing for a part of instances [20]. However, such methods still rely on

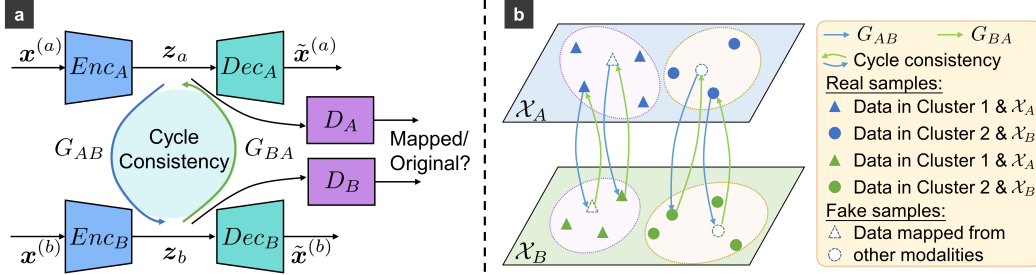

Figure 1: Overview of the proposed method. (a) Our adversarial network architecture for the unified representation learning. (b) Cycle consistency across modality-specific latent spaces illustrated on some samples. The cross-modal mappings help unify all the samples into a space.

samples with complete view information to perform the feature alignment. For unpaired multi-view data, constrained multi-view clustering is presented as a solution with 'must link' or 'cannot link' constraints on instance pairs [36]. Yet the constraint itself is a kind of extra pairing information. Meanwhile, other pairwise constraints such as co-occurrence frequency are also adopted to guide the clustering [11, 22]. Different from these studies, we seek a clustering algorithm for those unpaired multi-modal data in this paper without any extra prior knowledge.

**Adversarial learning for unpaired data.** To overcome the situation that paired data are often difficult to collect, some task-specific adversarial networks are developed to learn common representations across different domains [28, 27]. Meanwhile, a general solution, cycle consistency, is adopted in CycleGAN [39] to regularize the structured data. The key idea lies in that the instinct transitivity that mapping the data from one domain to another and then back to the original domain facilitates the reconstruction of the original data. As an extension, Augmented CycleGAN [1] learns the many-to-many cross-modal mappings based on this property. The cycle consistency constraint is widely used in cross-domain tasks like domain adaptation [16], hand tracking [23], image dehazing [33], together with cross-modal tasks such as hashing [31], visual-audio mutual generation [15] and cross-modal image synthesis [37]. In these applications, there often exists other supervising information like identities or positions to help align the domains or modalities in the adversarial learning. Therefore, none of these approaches are directly available for the mixed-modal clustering task. Since we do not rely on information of this kind on our task, we merely adopt the cycle consistency as the regularizer for constructing the cross-modal translations in our framework.

## 3 Methodology

In this section, we first introduce the setting of mixed-modal clustering. Then we present the detailed description of our proposed framework to tackle this problem. For convenience, we discuss the problem of clustering for data in two mixed modalities. In fact, it is easy to extend the proposed method to deal with data in several mixed modalities.

### 3.1 Problem setting

Given a set of $n$ mixed-modal samples $\mathcal{D} = \{\boldsymbol{x}_i\}_{i=1}^n$, where each sample is from either modality $A$ or $B$, our objective is to learn unified representations for the two modalities and then group the samples into $k$ categories. Obviously $\mathcal{D}$ can be divided into two single-modal sets $\mathcal{D}_A = \{\boldsymbol{x}_i^{(a)}\}_{i=1}^{n_a}$ and $\mathcal{D}_B = \{\boldsymbol{x}_i^{(b)}\}_{i=1}^{n_b}$, where $n = n_a + n_b$. As depicted in Figure 2, in traditional multi-modal/view clustering tasks, $\boldsymbol{x}_i^{(a)}$ and $\boldsymbol{x}_i^{(b)}$ both characterize the same instance. In contrast, $\boldsymbol{x}_i^{(a)}$ and $\boldsymbol{x}_i^{(b)}$ represent different instances in the mixed-modal setting. Namely, we do not have paired

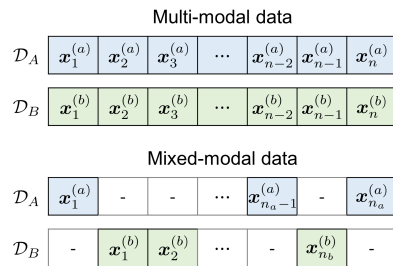

Figure 2: Comparison between multi-modal and mixed-modal data with two modalities.

samples from both modalities to uncover the correlation across modalities. Such a setting may occur in the case that, we want to find how many groups of topics the images and texts extracted from correlated web pages, which naturally form a mixed-modal dataset, could fall into.

## 3.2 Deep Mixed-Modal Clustering

The key problem of mixed-modal clustering is to unify the representations for mixed-modal data in the absence of pairing information. Motivated by the way that CycleGAN deals with unpaired data, we implement a similar adversarial network to learn the unified representations for mixed-modal clustering. The network architecture is illustrated in Figure 1(a). First, the divided single-modal data are transformed onto latent spaces and then recovered to the original spaces by corresponding auto-encoders. In the latent spaces, data are then mapped between the modality spaces via two cross-modal mappings. Meanwhile, two classifiers are adopted to identify whether a representation is mapped to or originally lies in the specific modality space. In the following, we describe the roles for these modules in our method.

**Latent representations.**   Before modeling the correlation across modalities, we first learn latent representations for each modality individually. In an unsupervised manner, it is easy to obtain the latent embeddings via two modality-specific auto-encoders:

$$
\begin{aligned}
\mathcal{L}_{\text{rec}}^{\text{A}}(\boldsymbol{\Theta}_{AE_A}) &= \|\boldsymbol{x}_i^{(a)} - Dec_A(Enc_A(\boldsymbol{x}_i^{(a)}))\|_2^2, \\
\mathcal{L}_{\text{rec}}^{\text{B}}(\boldsymbol{\Theta}_{AE_B}) &= \|\boldsymbol{x}_i^{(b)} - Dec_B(Enc_B(\boldsymbol{x}_i^{(b)}))\|_2^2,
\end{aligned}
\tag{1}
$$

where $Enc_*$ and $Dec_*$ denote the encoder and decoder for one modality respectively, and $\boldsymbol{\Theta}_{AE_A}$ and $\boldsymbol{\Theta}_{AE_B}$ are the parameter sets for each auto-encoder.

**Unpaired cross-modal mappings.**   In the latent spaces of $A$ and $B$, since learning a joint semantic space is not available, we turn to build the bidirectional mappings across $A$ and $B$, *i.e.*, $G_{AB} : \mathcal{X}_A \mapsto \mathcal{X}_B$ and $G_{BA} : \mathcal{X}_B \mapsto \mathcal{X}_A$. Though it is hard to directly constrain the individual modality mappings, we notice that the learned mapping functions should obey a *cycle-consistent* rule that, mapping samples lying in one modality space to another and then back to the original space should produce the original samples. This property intuitively provides us with a means of jointly constraining these cross-modal mappings to preserve correct modal information for samples via closed-loop reconstructions. In other words, with the learned latent codes $\boldsymbol{z}_a = Enc_A(\boldsymbol{x}^{(a)})$ and $\boldsymbol{z}_b = Enc_B(\boldsymbol{x}^{(b)})$, we build the mapping functions by pursuing $G_{BA}(G_{AB}(\boldsymbol{z}_a)) \approx \boldsymbol{z}_a$ and $G_{AB}(G_{BA}(\boldsymbol{z}_b)) \approx \boldsymbol{z}_b$, as shown in Figure 1(b). Let $\boldsymbol{\Theta}_{G_{AB}}$ and $\boldsymbol{\Theta}_{G_{BA}}$ be the parameters of $G_{AB}$ and $G_{BA}$, respectively. Using $\ell_1$ norm as the penalty, the cycle-consistency can be expressed as follows:

$$
\begin{aligned}
\mathcal{L}_{\text{cyc}}^{\text{A}}(\boldsymbol{\Theta}_{G_{AB}}, \boldsymbol{\Theta}_{G_{BA}}) &= \mathbb{E}_{\boldsymbol{z}_a \sim \mathcal{X}_A}\left[\|\boldsymbol{z}_a - G_{BA}(G_{AB}(\boldsymbol{z}_a))\|_1\right], \\
\mathcal{L}_{\text{cyc}}^{\text{B}}(\boldsymbol{\Theta}_{G_{AB}}, \boldsymbol{\Theta}_{G_{BA}}) &= \mathbb{E}_{\boldsymbol{z}_b \sim \mathcal{X}_B}\left[\|\boldsymbol{z}_b - G_{AB}(G_{BA}(\boldsymbol{z}_b))\|_1\right].
\end{aligned}
\tag{2}
$$

In another respect, the mapping functions could be interpreted as a special inter-modal auto-encoder. For modality *A*, this auto-encoder transforms the data points into another modality space *B* via the encoder $G_{AB}$ and reconstructs the data in *A* via the decoder $G_{BA}$. Meanwhile, the roles of encoder and decoder exchange in terms of modality *B*. Unlike traditional auto-encoders, this network utilizes the $\ell_1$ norm to pursue a sparse reconstruction error. The cycle consistency property requires this inter-modal auto-encoder to build reasonable translations across the two modality spaces. When $\mathcal{L}_{\text{cyc}}^{\text{A}} \to 0$, $\mathcal{L}_{\text{cyc}}^{\text{B}} \to 0$, we could recover the condition that $G_{AB} \circ G_{BA}(\cdot) = G_{BA} \circ G_{AB}(\cdot) = I(\cdot)$.

**Adversarial learning.**   To further improve the learning of mapping functions, we introduce the adversarial learning scheme into our framework. In this scheme, the mapping functions $G_{BA}$ and $G_{AB}$ are naturally viewed as generators producing fake samples that are transformed from other modalities rather than originally lying in a specific modality space. Meanwhile, we implement two discriminators $D_A$ and $D_B$ as their adversaries respectively to distinguish whether a sample is mapped from other modality spaces. Specifically, for modality *A*, the generator $G_{BA}$ and the discriminator $D_A$ play a mini-max game that, $D_A$ attempts to discriminate the real samples $\boldsymbol{z}_a$ from fake samples $G_{BA}(\boldsymbol{z}_b)$, while $G_{BA}$ aims at fooling $D_A$ by minimizing the difference between real and fake samples. Analogously, there is a similar gaming process between $D_B$ and $G_{AB}$. When the games reach the equilibrium, it is expected that both mappings could fit the distribution of real

---

**Algorithm 1** `Deep mixed-modal clustering algorithm`

---

**Input:** Mixed-modal dataset $\mathcal{D} = \{\boldsymbol{x}_i\}_{i=1}^n$, learning rate $\alpha$, weight range $c$, hyper-parameter $\lambda_1, \lambda_2$
**Output:** Parameter set $\boldsymbol{\Theta}$, clustering labels $\boldsymbol{y} = [y_1, \cdots, y_n]$

1: Pre-train auto-encoders $\boldsymbol{\Theta}_{AE_A}$ and $\boldsymbol{\Theta}_{AE_B}$ with corresponding single-modal data
2: **while** $\boldsymbol{\Theta}_{G_{AB}}$ and $\boldsymbol{\Theta}_{G_{BA}}$ not converged **do**
3:      Sample a batch of data from $\mathcal{D}_A$ and a batch of data from $\mathcal{D}_B$
4:      Update auto-encoders $\boldsymbol{\Theta}_{AE_A}$ and $\boldsymbol{\Theta}_{AE_B}$ with Eq. (2)          ▷ Auto-encoders
5:      **for** $t \leftarrow 1$ **to** $n\_critics$ **do**
6:          Sample a batch of data from $\mathcal{X}_A$ and a batch of data from $\mathcal{X}_B$
7:          $\boldsymbol{\Theta}_{D_A} \leftarrow \text{RMSprop}(\boldsymbol{\Theta}_{D_A}, -\nabla_{\boldsymbol{\Theta}_{D_A}} \mathcal{L}_{\text{adv}}^{\text{A}}, \alpha)$          ▷ Discriminators
8:          $\boldsymbol{\Theta}_{D_B} \leftarrow \text{RMSprop}(\boldsymbol{\Theta}_{D_B}, -\nabla_{\boldsymbol{\Theta}_{D_B}} \mathcal{L}_{\text{adv}}^{\text{B}}, \alpha)$
9:          $\boldsymbol{\Theta}_{D_A} \leftarrow \text{clip}(\boldsymbol{\Theta}_{D_A}, -c, c)$
10:         $\boldsymbol{\Theta}_{D_B} \leftarrow \text{clip}(\boldsymbol{\Theta}_{D_B}, -c, c)$
11:      **end for**
12:     Sample a batch of data from $\mathcal{X}_A$ and a batch of data from $\mathcal{X}_B$
13:     $\boldsymbol{\Theta}_{G_{AB}} \leftarrow \text{RMSprop}(\boldsymbol{\Theta}_{G_{AB}}, \nabla_{\boldsymbol{\Theta}_{G_{AB}}}(\mathcal{L}_{\text{adv}}^{\text{A}} + \lambda_1 \mathcal{L}_{\text{cyc}}^{\text{A}} + \lambda_1 \mathcal{L}_{\text{cyc}}^{\text{B}}), \alpha)$      ▷ Generators
14:     $\boldsymbol{\Theta}_{G_{BA}} \leftarrow \text{RMSprop}(\boldsymbol{\Theta}_{G_{BA}}, \nabla_{\boldsymbol{\Theta}_{G_{BA}}}(\mathcal{L}_{\text{adv}}^{\text{B}} + \lambda_1 \mathcal{L}_{\text{cyc}}^{\text{A}} + \lambda_1 \mathcal{L}_{\text{cyc}}^{\text{B}}), \alpha)$
15: **end while**
16: Transform all the data into a modality space as embedding $\boldsymbol{Z}$
17: $\boldsymbol{y} \leftarrow \text{Clustering}(\boldsymbol{Z})$

---

samples so that discriminators are confused. The resulted transformations, uncovering the correlation across two modalities, enable us to build unified representations for samples. Here we adopt the popular Wasserstein adversarial loss [3] as the objective function:

$$
\begin{aligned}
\mathcal{L}_{\text{adv}}^{\text{A}}(\boldsymbol{\Theta}_{G_{BA}}, \boldsymbol{\Theta}_{D_A}) &= \mathbb{E}_{\boldsymbol{z}_a \sim \mathcal{X}_A}[D_A(\boldsymbol{z}_a)] - \mathbb{E}_{\boldsymbol{z}_b \sim \mathcal{X}_B}[D_A(G_{BA}(\boldsymbol{z}_b))], \\
\mathcal{L}_{\text{adv}}^{\text{B}}(\boldsymbol{\Theta}_{G_{AB}}, \boldsymbol{\Theta}_{D_B}) &= \mathbb{E}_{\boldsymbol{z}_b \sim \mathcal{X}_B}[D_B(\boldsymbol{z}_b)] - \mathbb{E}_{\boldsymbol{z}_a \sim \mathcal{X}_A}[D_B(G_{AB}(\boldsymbol{z}_a))].
\end{aligned}
\tag{3}
$$

As a requirement in this loss, $D_A$ and $D_B$ should be 1-Lipschitz continuous. Therefore an 1-Lipschitz constraint is also enforced on them following $\|D_A(\boldsymbol{x}_1) - D_A(\boldsymbol{x}_2)\|_2 \leq \|\boldsymbol{x}_1 - \boldsymbol{x}_2\|_2$ and $\|D_B(\boldsymbol{x}_1) - D_B(\boldsymbol{x}_2)\|_2 \leq \|\boldsymbol{x}_1 - \boldsymbol{x}_2\|_2$. Following the strategy in [3], we adopt weight clipping for the discriminators to enforce this property.

**Overall objective.** With all these modules defined, we denote the whole parameter set as $\boldsymbol{\Theta} = \{\boldsymbol{\Theta}_{G_{AB}}, \boldsymbol{\Theta}_{G_{BA}}, \boldsymbol{\Theta}_{D_A}, \boldsymbol{\Theta}_{D_B}, \boldsymbol{\Theta}_{AE_A}, \boldsymbol{\Theta}_{AE_B}\}$. Putting all these together, our final objective function can be formulated as:

$$
\mathcal{L}(\boldsymbol{\Theta}) = \mathcal{L}_{\text{adv}}^{\text{A}} + \mathcal{L}_{\text{adv}}^{\text{B}} + \lambda_1(\mathcal{L}_{\text{cyc}}^{\text{A}} + \mathcal{L}_{\text{cyc}}^{\text{B}}) + \lambda_2(\mathcal{L}_{\text{rec}}^{\text{A}} + \mathcal{L}_{\text{rec}}^{\text{B}})
\tag{4}
$$

where $\lambda_1$ and $\lambda_2$ are the trade-off hyperparameters for cycle consistency and data reconstruction, respectively.

This corresponds to the following optimization problem:

$$
\min_{\substack{\boldsymbol{\Theta}_{G_{AB}}, \boldsymbol{\Theta}_{G_{BA}} \\ \boldsymbol{\Theta}_{AE_A}, \boldsymbol{\Theta}_{AE_B}}} \max_{\boldsymbol{\Theta}_{D_A}, \boldsymbol{\Theta}_{D_B}} \mathcal{L}(\boldsymbol{\Theta})
\tag{5}
$$

The learning procedure for our proposed framework is presented in Algorithm 1. First, we pretrain the auto-encoders individually to transform single-modal data onto the modality-specific latent spaces (line 1). Then in the loop, we alternatively update the auto-encoders (line 4), the discriminators (line 7–10) and generators (line 13–14) using RMSprop [29] to play the mini-max game. After the adversarial learning, we transform all the data points onto a modality space via the learned cross-modal mappings (line 16), *e.g.*, mapping the data in modality *B* onto *A*. See Figure 1(b). Then the unified latent representations are fed into a common clustering algorithm like $k$-means [4], by which we could obtain the final clustering results (line 17).

**Connection to optimal transport.** The Wasserstein loss is derived from the dual form of Wasserstein-1 distance through the Kantorovich-Rubinstein duality [26]. The Wasserstein-1 distance

for two distributions $\mathbb{P}_r$ and $\mathbb{P}_g$ is defined as:

$$W_1(\mathbb{P}_r, \mathbb{P}_g) = \inf_{\boldsymbol{\gamma} \in \prod(\mathbb{P}_r, \mathbb{P}_g)} \mathbb{E}_{(\boldsymbol{x}, \boldsymbol{y}) \sim \boldsymbol{\gamma}}[\|\boldsymbol{x} - \boldsymbol{y}\|_2] \tag{6}$$

where $\prod(\mathbb{P}_r, \mathbb{P}_g)$ is the set of all joint distributions with marginals $\mathbb{P}_r$ and $\mathbb{P}_g$. In fact, this is a special case of the minimal cost of an optimal transport problem [26], which aims at finding a plan $\boldsymbol{\gamma}(\boldsymbol{x}, \boldsymbol{y})$ to transport the mass from $\mathbb{P}_r$ to $\mathbb{P}_g$ at minimal cost. Here the cost moving one unit of mass from location $\boldsymbol{x}$ to location $\boldsymbol{y}$ is measured by the $\ell_2$ distance between the two points. And the transport plan $\boldsymbol{\gamma}(\boldsymbol{x}, \boldsymbol{y})$ could be intuitively interpreted as the mass that must be transported from location $\boldsymbol{x}$ to $\boldsymbol{y}$ in order to transform the distribution from $\mathbb{P}_r$ to $\mathbb{P}_g$. In our method, the mini-max game for $\mathcal{L}_{\mathrm{adv}}^{\mathrm{A}}$ could be viewed as solving the optimal transport problem between the distribution $\mathcal{X}_A$ and the distribution of $G_{BA}(\boldsymbol{z}_b)$ with $\boldsymbol{z}_b$ a random variable with density $\mathcal{X}_B$. That is, the target of this game is to find an optimal transport map from $\mathcal{X}_B$ to $\mathcal{X}_A$, which is precisely the generator $G_{BA} : \mathcal{X}_B \mapsto \mathcal{X}_A$ in this problem. Likewise, the mini-max game for $\mathcal{L}_{\mathrm{adv}}^{\mathrm{B}}$ could learn the optimal transportation plan $G_{AB} : \mathcal{X}_A \mapsto \mathcal{X}_B$.

# 4 Experiments

In this section, we provide the empirical evaluation on two real-world mixed-modal dataset, *Wikipedia* and *NUS-WIDE-10K*.

**Competitors.** We compare our framework with the following clustering methods including a classical model and several recent deep methods:

(1) $k$-means [4]: As a classical clustering algorithm, $k$-means proceeds by alternating between the cluster assignment and the centroid update steps.

(2) DCN [32]: The network integrates $k$-means module into an auto-encoder, thus could jointly learning clustering and representations.

(3) DKM [9]: Unlike DCN which alternatively updates network and clustering parameters, DKM reformulates the problem so that the whole framework could be jointly optimized by gradient-based solvers.

(4) IDEC [14]: IDEC also incorporates an auto-encoder with a clustering loss in the latent space, which guides the learning of centroids via measuring the difference between teacher and target distributions.

(5) IMSAT [17]: This method learns discrete data representations and performs clustering in an end-to-end fashion through combining self-augmented training and information-theoretic dependency maximizing between learned codes and original data.

**Evaluation metrics.** In the experiment, we measure the performance using five classic clustering metrics [2]: Clustering Accuracy, Adjusted Rand Index (ARI), Normalized Mutual Information (NMI), F-score, and Purity. They measure the quality of clustering from different perspective. For the predicted and ground-truth labels, ARI measures their similarity through pairwise comparisons; NMI measures their agreement by considering the disorder of clusters; Purity calculates how they matched based on the predicted label frequency. F-score is the harmonic mean of clustering precision and recall. Note that all these metrics ignore the permutations of cluster labels except for accuracy, thus a best mapping using Hungarian algorithm [19] should be computed between cluster and ground-truth labels before calculating the accuracy. For all the metrics, the value 1 means a perfect clustering.

**Settings.** All the experiments are performed on Ubuntu 16.04 with a NVIDIA GTX 1080 Ti GPU. Our proposed method is implemented using PyTorch 1.0 [24]. For the clustering process of the proposed method, we choose the modality whose data are more informative as the final modality to be transformed into. Unfortunately, on both datasets used in our paper, deep features are available for image modality (*A*), while the text modality (B) only contains binary features. In this way, the latent representations learned for *B* obviously have less representability than those for *A*. As a result, we transform all the data points into modality *A* in our experiments.

Table 2: Dataset statistics.

| Dataset | Modality1 | Modality2 | Training samples | Test samples | Categories |
|---|---|---|---|---|---|
| Wikipedia | image | text (article) | 1910 | 256 | 10 |
| NUS-WIDE-10K | image | text (tag) | 7500 | 2500 | 10 |

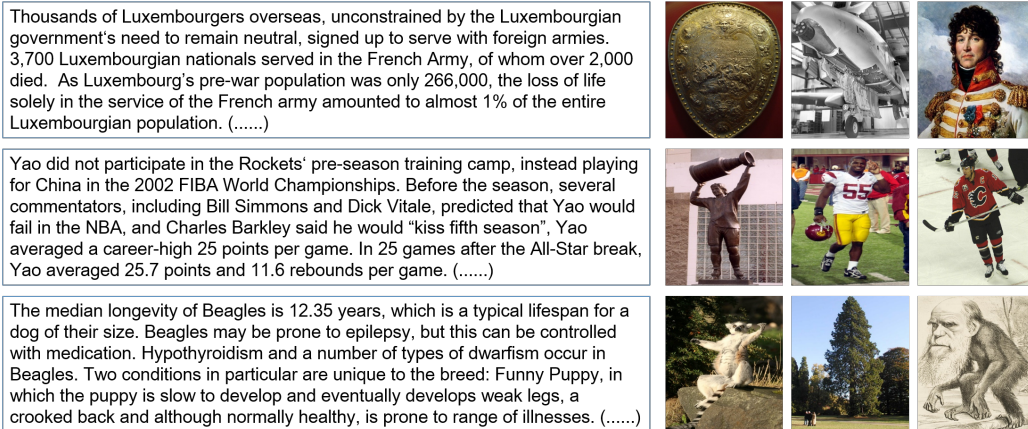

Figure 3: Mixed-modal examples on Wikipedia dataset. In each row, the text and images belong to the same semantic category. From top to bottom, the three categories are *warfare, sport* and *biology*, respectively.

## 4.1 Wikipedia dataset

**Dataset description.** The *Wikipedia* dataset[2] [25] contains 2,866 image-text pairs selected from the Wikipedia's "featured articles" collection. The text in each pair is a paragraph describing the content of the corresponding image. According to the collection, these pairs are divided into 10 semantic categories. For each pair, we select the image or the text as the sample uniformly at random and discard the other modality, leading to a dataset with mixed modalities. Then we choose 30% of the samples as test set. Consequently, there are 1910/956 samples in train/test set. The statistics of this dataset are shown in Table 2. Meanwhile, we present some examples on the resulted dataset in Figure 3. In the real-world scenario, we will frequently face such data, in which data with multiple modalities are mixed up during the collection process.

**Implementation details.** Instead of implementing neural networks to learn latent representations for the raw data, we simply reconstruct extracted features to better focus on the unsupervised learning for the cross-modal mappings. Therefore, we use the 4096d vector extracted by the second last layer of VGG-Net as the initial image representation and 5000d bag-of-words vector as the initial text representation, which are both provided in [30]. To make the evaluation available for the competitors, we perform PCA on those features for each modality to reduce them into the same dimension of 2048. The preprocessed features are also used to our model for fair comparison. Then we adopt two deep auto-encoders with the following fully-connected structure: $1024 \to 256 \to 128 \to 256 \to 1024$. The generators are built with $128 \to 256 \to 128$ and the discriminators are $32 \to 1$. For the auto-encoders and generators, each fully-connected layer is followed with a batch normalization layer and a LeakyReLU layer with the negative slope 0.2. According to the architecture, we empirically set the learning rates for the auto-encoders, generators and discriminators to 1e-3, 1e-4, 5e-5, respectively. Meanwhile, the trade-off coefficient $\lambda_1$ is set to 1 and $\lambda_2$ is set to 2 for the objective function. For the weight clipping, the clipping range is fixed at 0.05.

**Results.** The experimental results for all the involved models are depicted in Table 3. It is shown that our method outperforms the competitors over Accuracy, NMI, F-score and Purity, and achieves the second highest performance in terms of ARI and Precision at a slight disadvantage. Specifically,

Table 3: Performance comparisons on Wikipedia dataset. The larger the better.

| Algorithm | Accuracy | ARI | NMI | F-score | Purity |
|-----------|----------|--------|--------|---------|--------|
| $k$-means | 0.2291 | 0.0166 | 0.1003 | 0.1857 | 0.2301 |
| DKM | 0.2173 | 0.0108 | 0.1170 | 0.1729 | 0.2429 |
| DCN | 0.2215 | 0.0137 | 0.1172 | 0.1688 | 0.2465 |
| IDEC | 0.2153 | 0.0375 | 0.0849 | 0.1654 | 0.2606 |
| IMSAT | 0.2521 | 0.0573 | 0.1093 | 0.1738 | 0.2720 |
| Ours | 0.2720 | 0.0558 | 0.1543 | 0.1878 | 0.3075 |

our model outperforms the second best method by 0.0199, 0.0371, 0.0355 over Accuracy, NMI, and Purity. These results show the effectiveness of our proposed method in tackling mixed-modal clustering problem. Besides, we could make comprehensive observations as follows. (1) We see that $k$-means and the derived deep clustering methods DKM, DCN obtain similar clustering results, and $k$-means even achieves higher values on most metrics. This indicates that in the mixed-modal setting, introducing of non-linearity via deep neural networks into clustering models could not simply improve the performance. It is the most important to learn a unified representation for samples as we do in our proposed method. (2) Moreover, $k$-means, DKM and DCN all get much lower ARI values than others. These methods may perform like random label assignment in this setting. (3) Though IMSAT benefits from the information-theoretical regularization and thus obtains a good performance compared with other competitors, the proposed method still outperforms it over 5 metrics. This again justifies our method and the importance of unifying the modality-specific representations.

### 4.2 NUS-WIDE-10K

**Dataset description.** The *NUS-WIDE-10K* dataset[3] [10] consists of 10,000 image-text pairs evenly selected from the 10 largest semantic categories of *NUS-WIDE* dataset [8]. Namely, there are 1,000 pairs for each class. In this dataset, tags serve as the text modality. Likewise, we randomly select either the image or the text as our sample for each pair, then split the whole dataset into training set with 7500 samples and test set with 2500 samples. The statistics are displayed in Table 2.

**Implementation details.** Similar to Wikipedia dataset, we build auto-encoders for the 4096d image features extracted by VGG-Net and the 1000d bag-of-words text features provided in [30], respectively. Similarly, they are reduced to 1000d using PCA so that the mixed-modal data could be learned by the competitors. The structure of the auto-encoders is: $512 \rightarrow 256 \rightarrow 128 \rightarrow 256 \rightarrow 512$. The generators are build with $128 \rightarrow 128$ and the discriminators are $32 \rightarrow 1$. The learning rates for the auto-encoders, generators and discriminators are empirically set to 5e-4, 5e-5, 5e-5, respectively. $\lambda_1$ and $\lambda_2$ are both set to 1 to balance the loss. Moreover, the weight clipping range is fixed at 0.05 which is the same as in Wikipedia.

**Results.** All the quantitative results are summarized in Table 4. On this dataset, our method achieves better performance against the baselines on most. It is worth mentioning that the results of the proposed method are higher than the second best's with 0.0220, 0.0566, 0.0439, 0.0196 with regard to Accuracy, ARI, NMI, and Purity. Different from the results on Wikipedia, all the competitors achieve quite low ARI or NMI values, and even some of them are very close to 0. Such performance degradation may come from two aspects. One is that PCA performed on the image features may result in some loss of information. The other is that features extracted from tags are much simple, and the semantic gap between image and text modality space is much larger than on Wikipedia. However, our method still obtains a relatively much higher ARI and NMI values than these models. This again demonstrates the effectiveness of our unified representation learning via cycle-consistent mappings.

Table 4: Performance comparisons on NUS-WIDE-10K dataset. The larger the better.

| Algorithm | Accuracy | ARI | NMI | F-score | Purity |
|---|---|---|---|---|---|
| $k$-means | 0.2744 | 0.0044 | 0.0469 | 0.3008 | 0.5208 |
| DKM | 0.2932 | 0.0130 | 0.0116 | 0.2901 | 0.5036 |
| DCN | 0.3036 | 0.0144 | 0.0512 | 0.2959 | 0.5296 |
| IDEC | 0.3045 | 0.0006 | 0.0082 | 0.3048 | 0.5036 |
| IMSAT | 0.3080 | 0.0038 | 0.0064 | 0.3422 | 0.5036 |
| Ours | 0.3300 | 0.0710 | 0.0951 | 0.3043 | 0.5492 |

## 4.3 Ablation Study

To see how much the adversarial training and modality unifying contribute to our model, we further conduct the ablation experiments on both datasets. As the ablated variants, latent modality-specific representations obtained before/after the adversarial training are fed into $k$-means for evaluation. The results are recorded in Table 5. We can observe that the performance of our model is largely improved by the final cross-modal transformations. Without utilizing the transformations, our model could only obtain a similar performance as $k$-means. This indicates that the unifying of modality-specific representations could reduce the semantic gap between the modalities.

Table 5: Ablation study on Wikipedia and NUS-WIDE-10K. *adv.* denotes the adversarial learning, and *uni.* means the final modality unifying using the learned cross-modal translation.

| Dataset | adv. | uni. | Accuracy | ARI | NMI | F-score | Purity |
|---|---|---|---|---|---|---|---|
| Wikipedia | | | 0.2301 | 0.0340 | 0.1069 | 0.1730 | 0.2563 |
| | ✓ | | 0.2395 | 0.0290 | 0.1311 | 0.1696 | 0.2699 |
| | ✓ | ✓ | 0.2720 | 0.0558 | 0.1543 | 0.1878 | 0.3075 |
| NUS-WIDE-10K | | | 0.2696 | 0.0321 | 0.0719 | 0.2323 | 0.5332 |
| | ✓ | | 0.2884 | 0.0359 | 0.0672 | 0.2542 | 0.5336 |
| | ✓ | ✓ | 0.3300 | 0.0710 | 0.0951 | 0.3043 | 0.5492 |

## 5 Conclusion

In this paper, we make an early attempt to tackle the clustering task for mixed-modal data, where each sample is only characterized by one of several modalities. Inspired by CycleGAN, our proposed method unifies the modality-specific representations through learning the cycle-consistent mappings across modalities in an adversarial manner. Subsequently, our method performs a common clustering with the unified representations. We experimentally validate our model on two real-world datasets, where our model consistently outperforms the classical and deep clustering approaches over most metrics. The results also demonstrate the importance to adopt the cross-modal mappings for the mixed-modal setting. In the future, we plan to incorporate the optimal transport theory to solve this problem with theoretical guarantee and reach a better performance.

### Acknowledgments

This work was supported in part by the National Key R&D Program of China (Grant No. 2016YFB0800603), in part by National Natural Science Foundation of China: 61620106009, U1636214, 61836002, U1636214, 61733007, 61672514 and 61976202, in part by National Basic Research Program of China (973 Program): 2015CB351800, in part by Key Research Program of Frontier Sciences, CAS: QYZDJ-SSW-SYS013, in part by the Strategic Priority Research Program of Chinese Academy of Sciences, Grant No. XDB28000000, in part by Beijing Natural Science Foundation (No. KZ201910005007 and 4182079), in part by Peng Cheng Laboratory Project of Guangdong Province PCL2018KP004, and in part by Youth Innovation Promotion Association CAS.

## Footnotes

[2] http://www.svcl.ucsd.edu/projects/crossmodal/

[3] https://lms.comp.nus.edu.sg/research/NUS-WIDE.htm

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
