[Reviews · NeurIPS 2019]

Reviewer 1



The authors tackle the clustering problem in the setting where each sample is only characterized by one of multiple modalities. They present an adversarial deep model with cycle consistency concerned to learn translations across modal spaces for unifying the representations. This model is evaluated on two datasets and shows the improvement over baselines. Pros: - The paper is well-organized, especially the way the authors motivate the problem and present the solution. - The key problem being solved is novel and interesting to me. It seems to be the first work trying to deal with totally unpaired ‘multi-modal’ data without any other constraints to my best of knowledge. - The proposed method is capable of improving the clustering performance supported by empirical results. And, it seems to be a fresh perspective to interpret the learned cross-modal translations as the optimal transport plan between modal spaces. Cons: - Further discussions about the proposed model should be made to complete my understanding, e.g., the relationship between generators. - Some technical details are not very clear or intuitive. Typos: - Line 139: it may be more precise to use ‘parameter sets’ instead of ‘parameters’. - Line 157: there should be a ‘the’ before ‘discriminators’.

Reviewer 2



This paper proposes a novel and challenging task that clusters data in multiple modalities without any pairing information, with a simple and reasonably workable approach to solve it. The idea of cycle consistency is not novel since it has been already shown to be effective in addressing data with multiple modalities in the supervised setting. However, this work is the first trial to use this idea to solve the unpaired data clustering problem. Moreover, the authors demonstrate the connection between the proposed method and optimal transport, which may inspire the future work. In general, this paper is well written and easy to follow. The experimental results seem good. But the experiments are not that thoroughly executed for me. Some details should be elaborated and ablation study should be carried out. As an application paper, for me, its most valuable part lies in the task it proposes. As I mentioned in the former part, our community could benefit from such a novel task which will bring forth various innovative work. Typos: In Eq.(1), it seems that the summation symbol is missed. In line 170, ‘must be transport’ should be ‘must be transported’. In line 179, ‘are feed’ should be ‘are fed’.

Reviewer 3



his paper aims at solving the clustering problem on mixed-modal data. Unlike traditional settings, the modalities involved in this paper are represented in the total absence of pairing information, thus are very hard to be aligned. So it is very natural to learn the transformation functions between modalities based on the cycle consistency principle. Moreover, the experimental results indicate that this idea indeed improves the unification of mixed-modal data representations and bridges the semantic gap. Overall, the presentation of this paper is detailed. However, some of the statements are not very clear so they should be re-organized. Also, there are few typos to be corrected. 1. Line 119, divided -> split. 2. Line 129, set -> sets.

[Author Response · NeurIPS 2019]

# Response for Submission 3163 "DM2C: Deep Mixed-Modal Clustering"

We thank all the reviewers for their careful and valuable comments.

**To R1**

**Q1. Ablation study** We evaluate $k$-means using latent modality-specific representations obtained before/after the adversarial training (denoted as *modal-spec (with adv.)* and *modal-spec (w/o adv.)* respectively) in our model on the Wikipedia and NUS-WIDE-10K datasets. The results are recorded in Tab. S1. We can observe that the performance of our model is largely improved by the final cross-modal transformations. This indicates that the unification of modality-specific representations could reduce the semantic gap between the modalities. We will add the experiments and discussions if accepted.

Table S1: Ablation study on the Wikipedia and NUS-WIDE-10K datasets. The larger the better.

| Dataset | Algorithm | Accuracy | ARI | NMI | F-score | Precision | Recall | Purity |
|---------|-----------|----------|-----|-----|---------|-----------|--------|--------|
| Wikipedia | modal-spec (w/o adv.) | 0.2301 | 0.0340 | 0.1069 | 0.1730 | 0.1289 | 0.2633 | 0.2563 |
| | modal-spec (with adv.) | 0.2395 | 0.0290 | 0.1311 | 0.1696 | 0.1256 | 0.2611 | 0.2699 |
| | ours | **0.2720** | **0.0558** | **0.1543** | **0.1878** | **0.1439** | **0.2700** | **0.3075** |
| NUS-WIDE | modal-spec (w/o adv.) | 0.2696 | 0.0321 | 0.0719 | 0.2323 | 0.3318 | 0.1787 | 0.5332 |
| | modal-spec (with adv.) | 0.2884 | 0.0359 | 0.0672 | 0.2542 | 0.3316 | 0.2060 | 0.5336 |
| | ours | **0.3300** | **0.0710** | **0.0951** | **0.3043** | **0.3579** | **0.2648** | **0.5492** |

**Q2. The value of $n_{\text{critics}}$**: This value is empirically set to 5 in the experiments.

**To R2**

**Q1. Cycle consistency on multiple modalities**: Perhaps due to our way of writing, it is a pity to leave you an impression that only cross-domain cycle consistency is mentioned in related work. In fact, the reference [29] in our paper is an application of cross-modal cycle consistency. To the best of our knowledge there is only few other relevant work on cross-modal cycle consistency, be it "A Uniform Framework for Cross-Modal Visual-Audio Mutual Generation" (AAAI18) and "Multi-modal Cycle-consistent Generalized Zero-Shot Learning" (ECCV18). However none of them are directly available for the mixed-modal clustering task posed in our paper. We will add a detailed discussion on this issue if accepted.

**Q2. "1-Lipschitz constraint" is not explained**: "1-Lipschitz constraint" is a requirement of the dual formulation of the $\mathcal{W}_1$ distance. More exactly, in our case, it refers to the fact that $D_A(\cdot)$ and $D_B(\cdot)$ are 1-Lipschitz continuous, which means $\|D_A(\boldsymbol{x}_1) - D_A(\boldsymbol{x}_2)\| \leq \|\boldsymbol{x}_1 - \boldsymbol{x}_2\|$ and $\|D_B(\boldsymbol{x}_1) - D_B(\boldsymbol{x}_2)\| \leq \|\boldsymbol{x}_1 - \boldsymbol{x}_2\|$ (as what is put down in line 161-162).

**Q3. How to sample data from $\mathcal{X}_{A/B}$**: Directly sampling data from $\mathcal{X}_{A/B}$ requires transforming all the data points from $\mathcal{D}_{A/B}$ into $\mathcal{X}_{A/B}$ before sampling, which is impractical for high dimensional data. Hence we adopt a much simpler way that, we 1) sample a batch of data from $\mathcal{D}_{A/B}$, 2) feed them into the auto-encoder A/B to obtain the latent representations lying in $\mathcal{X}_{A/B}$.

**To R3**

**Q1. The relationship between generators**: Ideally, as you have suggested, the generators should satisfy $G_{AB} \circ G_{BA}(\cdot) = G_{BA} \circ G_{AB}(\cdot) = I(\cdot)$ considering the cycle consistency constraint. The cycle-consistency regularization terms $\mathcal{L}_{\text{cyc}}^{A}(\boldsymbol{\Theta}_{G_{AB}}, \boldsymbol{\Theta}_{G_{BA}})$, $\mathcal{L}_{\text{cyc}}^{B}(\boldsymbol{\Theta}_{G_{AB}}, \boldsymbol{\Theta}_{G_{BA}})$ guarantee that $\mathbb{E}_{\boldsymbol{z}_a \sim \mathcal{X}_A}[\|\boldsymbol{z}_a - G_{BA}(G_{AB}(\boldsymbol{z}_a))\|_1]$ and $\mathbb{E}_{\boldsymbol{z}_b \sim \mathcal{X}_B}[\|\boldsymbol{z}_b - G_{AB}(G_{BA}(\boldsymbol{z}_b))\|_1]$ are small. This approximates the cycle-consistency condition. When $\mathcal{L}_{\text{cyc}}^{A}(\boldsymbol{\Theta}_{G_{AB}}, \boldsymbol{\Theta}_{G_{BA}}) \to 0$, $\mathcal{L}_{\text{cyc}}^{B}(\boldsymbol{\Theta}_{G_{AB}}, \boldsymbol{\Theta}_{G_{BA}}) \to 0$, we exactly recover this condition.

**Q2. The choice of modality for the final clustering (only A is used in this paper)**: For the clustering process, we chose the modality whose data are more informative. In our setting, deep features are available for image modality (A), while the text modality (B) only contains binary features. In this way, the latent representations learned for B obviously have less representability than those for A. As a result, we transform all the data points into modality A.

[Meta-Review · NeurIPS 2019]

The authors introduce a new clustering problem, mixed-modal clustering, and presents a solution leveraging a novel application of cycle consistency and adversarial learning. The method is well-explained and supported by experimental results.